# AP39, a novel mitochondria-targeted hydrogen sulfide donor ameliorates doxorubicin-induced cardiotoxicity by regulating the AMPK/UCP2 pathway

**Bin Zhang, Yangxue Li, Ning Liu, Bin Liu** *

The Second Hospital of Jilin University, Nanguan District, Changchun City, Jilin Province, China

* liubin3333@vip.sina.com

## Abstract

Doxorubicin (DOX) is a broad-spectrum, highly effective antitumor agent; however, its cardiotoxicity has greatly limited its use. Hydrogen sulfide ($H_2S$) is an endogenous gaseous transmitter that exerts cardioprotective effects via the regulation of oxidative stress and apoptosis and maintenance of mitochondrial function, among other mechanisms. AP39 is a novel mitochondria-targeted $H_2S$ donor that, at appropriate concentrations, attenuates intracellular oxidative stress damage, maintains mitochondrial function, and ameliorates cardiomyocyte injury. In this study, DOX-induced cardiotoxicity models were established using H9c2 cells and Sprague–Dawley rats to evaluate the protective effect of AP39 and its mechanisms of action. Both in vivo and in vitro experiments showed that DOX induces oxidative stress injury, apoptosis, and mitochondrial damage in cardiomyocytes and decreases the expression of p-AMPK/AMPK and UCP2. All DOX-induced changes were attenuated by AP39 treatment. Furthermore, the protective effect of AP39 was significantly attenuated by the inhibition of AMPK and UCP2. The results suggest that AP39 ameliorates DOX-induced cardiotoxicity by regulating the expression of AMPK/UCP2.

## Introduction

Doxorubicin (DOX), a broad-spectrum anthracycline antineoplastic drug, is widely used for the treatment of leukemia, breast cancer, ovarian cancer, lymphoma, and osteosarcoma [1], and plays an extremely important role in tumor chemotherapy. However, DOX has dose-dependent, cumulative and progressive cardiotoxicity [2], which mainly manifests as arrhythmia, heart failure, myocardial injury, hypertension, and cardiomyopathy. Some studies have shown that the incidence of DOX-induced cardiotoxicity significantly increases with the increase in the total cumulative dose of DOX in a day or in a therapeutic cycle, and the mortality rate after treatment with a single or a cumulative dose of DOX of 5-25mg/kg is 10%-38%. After 2 years of chemotherapy, the mortality rate has been reported to reach 50% [3]. This not only limits the therapeutic dose of DOX, but also greatly affects the quality of life of cancer survivors and may even shorten their life expectancy. Studies have shown that DOX-induced

**Funding:** This work was supported by Jilin Province Science and Technology Department (202220303002SF), Jilin Provincial Development and Reform Commission (2022C003), Jilin Province Science and Technology Department (20190905002SF). The funders had no role in study design, data collection and analysis, decision to publish, or preparation of the manuscript.

**Competing interests:** The authors have declared that no competing interests exist.

cardiotoxicity is mainly associated with oxidative stress, lipid peroxidation damage, apoptosis and mitochondrial dysfunction. Furthermore, the effects are also reported with inducing inflammation and affecting DNA replication and transcription [4–7]. Currently, the only drug approved by the FDA for the treatment of DOX cardiotoxicity is dexrazoxane, which still has various side effects, including myelotoxicity in patients with soft-tissue sarcoma [8]. Therefore, there is an urgent need to develop or find more safe and effective drugs that may ameliorate DOX cardiotoxicity, explore their mechanisms of action, and translate these agents into clinical applications.

Hydrogen sulfide ($H_2S$), a gaseous signaling molecule, plays an important role in cardiovascular diseases by ameliorating atherosclerosis and hypertension [9], attenuating myocardial ischemia-reperfusion (I/R) injury [10,11], ameliorating heart failure [12,13], and attenuating myocardial fibrosis [14,15]. This may be related to its ability to regulate oxidative stress, apoptosis, autophagy, inflammation, mitochondrial function, neovascularization, and fibrosis at reasonable concentrations [16,17]. AP39 is a novel mitochondria-targeted $H_2S$ donor that attenuates intracellular oxidative stress at appropriate concentrations while maintaining cell viability, mitochondrial respiration, and mitochondrial DNA integrity [18,19]. It prevents myocardial ischemia-reperfusion injury independently of the cytoplasmic RISK pathway [20], inhibits mitochondrial autophagy, antagonizes cardiomyocyte iron death, and ameliorates myocardial fibrosis in rats with myocardial infarction via the PINK1/Parkin pathway [21]. Supplementation of preservation fluid with AP39 protects heart grafts from long-term ischemic damage and prevents I/R injury in heart transplantation [22].

Adenosine monophosphate-activated protein kinase (AMPK) is an important regulator of cellular energy homeostasis and mitochondrial homeostasis. The activation of AMPK modulates cellular metabolism, autophagy, apoptosis, and fibrosis [23]. Uncoupling protein 2 (UCP2) is located within the inner mitochondrial membrane and affects mitochondrial function and metabolism through oxidative phosphorylation uncoupling. AMPK attenuates oxidative stress damage, reduces apoptosis [24], attenuates mitochondrial damage [25], and attenuates inflammatory responses [26] by upregulating UCP2.

DOX causes cardiotoxicity through a variety of possible molecular mechanisms related to AMPK [27], including oxidative stress, mitochondrial damage, and apoptosis, whereas $H_2S$ increases mitochondrial ATP synthesis, induces mitochondrial biogenesis [28], ameliorates oxidative stress, and reduces apoptosis via AMPK [29]. Based on its effects on oxidative stress, apoptosis, and mitochondrial processes, we hypothesized that the exogenous $H_2S$ donor AP39 may attenuate DOX-induced cardiotoxicity. The aim of this study was to assess whether AP39 exerts a protective effect against DOX-induced cardiotoxicity and to investigate its mechanism of action, including its effects on the mitochondrial pathway and AMPK/UCP2.

## Materials and methods

### Reagents and antibodies

DOX (S1208) was purchased from Selleck (Houston, TX, USA), AP39(HY-126124) was purchased from MCE, Compound C (CC; 171260) and genipin (G4796) were purchased from Sigma–Aldrich (St. Louis, MO, USA). The following primary antibodies for the following proteins were purchased from Cell Signaling Technology (Danvers, MA, USA): Caspase-3(9662, 1:1000), Cleaved Caspase-3 (9664, 1:1000), AMPKα (5831, 1:1000), p-AMPKα (Thr172) (50081, 1:1000), and UCP2 (89326, 1:1000). Primary antibodies for Bax (A0207, 1:1000) and Bcl-2 (A19693, 1:1000) were purchased from ABclonal (Wuhan, China). Small interfering RNA against UCP2 (siUCP2) and its negative control (NC) were synthesized by IBSBIO (Shanghai, China). Lipofectamine 2000(11668500) was purchased from Invitrogen (Waltham, MA, USA). Annexin V-FITC (331200) and SYTOX Red (S34859) were purchased from

Thermo Fisher Scientific (Waltham, MA, USA). The Cell Counting Kit-8(CCK-8)(BA00208) and BCA Protein Assay Kit (C05-02001) were purchased from Bioss (Beijing, China). 2′,7′-dichlorofluorescein diacetate (DCFH-DA) (BB-47053) was purchased from Bestbio (Nanjing, China). The Mitochondrial Membrane Potential Assay Kit with JC-1 (J8030) and $H_2S$ Content Assay Kit (BC2055) were purchased from Solarbio (Beijing, China). The ATP assay kit (A095-1-1) was purchased from Nanjing Jiancheng Bioengineering Institute (Nanjing, China). The Superoxide Dismutase (SOD) Activity Assay Kit (AKAO001M), Glutathione Peroxidase (GPX) Activity Assay Kit (AKPR014M), Malondialdehyde (MDA) Content Assay Kit (AKFA013M), Coenzyme II NADP (H) Content Assay Kit (AKCO018M), and Lactate Dehydrogenase (LDH) Activity Assay Kit (AKCO003M) were purchased from Beijing Boxbio Science & Technology Co., Ltd. (Beijing, China). The Rat Troponin T Type 2, Cardiac (TNNT2) ELISA Kit (JL27509), Rat Creatine Kinase MB Isoenzyme (CKMB) ELISA Kit (JL12296), and Rat Brain Natriuretic Peptide (BNP) ELISA Kit (JL11495) were purchased from Jianglai Biology (Shanghai, China). The TUNEL Apoptosis Assay Kits (C1086 and C1091) were purchased from Beyotime (Shanghai, China).

## Animals and treatment

The study was approved by the Institutional Committee for the Protection and Utilization of Animals of Jilin University (Approval Number:2023 No.463). All handling of laboratory animals during experiments was in accordance with the Guidelines for the Management and Use of Laboratory Animals published by the National Institutes of Health. Animal studies were conducted in accordance with ARRIVE guidelines.

Male 8 to 10-week-old SPF Sprague–Dawley rats, weighing 300–320 g, were purchased from Yeast Laboratory Animal Technology. The rats were housed individually (one rat in one cage) at the Animal Center of Jilin University, with the room temperature and humidity controlled at $21 \pm 1°C$ and 50–60%, respectively, under a 12h day/night cycle. The animals had free access to water and food, and were acclimatized for 1 week. They were then randomly assigned to groups (10 rats per group) and administered treatments according to different protocols. The groups were as follows: (1) Con;(2) DOX;(3) AP39;(4) DOX+AP39;(5) DOX +AP39+Compound C;(6) DOX+AP39+genipin. The dosing was as follows: The Con group was administered an equal amount of 0.9% NaCl; DOX was administered intraperitoneally once a week for 3 weeks at a dose of 5 mg/kg, resulting in a cumulative dose of 15 mg/kg; AP39 was administered intraperitoneally once every other day at a dose of 50 nmol/kg, starting at the same time as DOX, for 3 weeks; Compound C (171260, Sigma, USA) was administered at a dose of 20mg/kg/d, one day prior to the start of DOX, for 1 week; and genipin was administered for 3 consecutive days prior to the initiation of DOX at a dose of 20 mg/kg/d. The above doses were based on previous study reports [30–33] and experimental data.

Weighing of the animals was performed every 3 days during the experiment, and their mental status, activity status, and any pain or discomfort were also monitored and recorded. The duration of this experiment was 21 days, and no rats died before euthanasia. After 21 days, cardiac ultrasound was performed after isoflurane anesthesia was administered, and then euthanasia was performed by $CO_2$ inhalation (a total of 60 rats). The above experiments were supervised and directed by the Institutional Committee for the Protection and Utilization of Animals of Jilin University, and all efforts were made to minimize suffering.

## Cell culture and treatments

The rat H9c2 cell (191377) was purchased from Beijing Zhongke QC Biotechnology Co. (Beijing, China). DMEM supplemented with 10% fetal bovine serum (Kang Yuan Biology, Tianjin,

China) and 1% penicillin and streptomycin (Solarbio, Beijing, China) was used for cell culture in an incubator at 37˚C and a 5% $CO_2$ atmosphere. Different drugs were given to stimulate the cells for 24h according to the experimental protocol including DOX (1 μmol/L), AP39 (100 nmol/L), and CC (10 μmol/L) [34]. To reduce UCP2 expression in vitro, cells were transfected with siUCP2 (50 nmol/L) using the transfection reagent Lipofectamine2000 for 48h, and the effectiveness of transfection was evaluated by qPCR and western blotting.

### Cell activity assay

H9c2 cells were inoculated in 96-well plates ($4 \times 10^3$/well) and incubated with different concentrations of DOX (0, 0.5, 1, and 2 μmol/L) and AP39 (0, 30, 50, 100, 300, and 500 nmol/L), with a final volume of 100μL in each well. After 24 h, 10 μL of CCK-8 reagent was added to each well, the cells were incubated in the cell incubator for 60 min, and absorbance was measured at 450 nm.

### Detection of ROS

H9c2 cells were inoculated in 6-well plates ($5 \times 10^4$/well), and different stimuli were applied when cells reached approximately 70% confluence. Cells were incubated for 24h in a cell culture incubator. The DCFH-DA probe was diluted with serum-free DMEM at a ratio of 1:1000, hoechst 33342 (C1029, Beyotime, China) at a ratio of 1:100, added to the 6-well plates at 1 mL/well, followed by incubation 37˚C in the dark for 20 min. Cells were washed gently with phosphate-buffered saline and images were obtained under a fluorescence microscope. The average fluorescence intensity was evaluated using ImageJ.

### Flow cytometry

H9c2 cells were resuspended under different conditions and diluted with 1× Binding Buffer to a concentration of $1 \times 10^6$ cells/mL. Then, 100 μL of the cell suspension was used for flow cytometry; briefly, 5 μL of Annexin V-FITC and 5 μL of SYTOX Red were added, samples were incubated at room temperature (25˚C) in the dark for 15 min, 400 μL of 1× Binding Buffer was added, and samples were assayed immediately using the flow cytometer (Cytoflex, Beckman).

### Western blotting

Total protein was extracted from cell samples and cardiac tissues using RIPA buffer, and the protein concentration was determined using a BCA Kit. Equal concentrations of protein samples were separated by 10% SDS-PAGE and then transferred to PVDF membranes (Millipore, USA), which were blocked with 5% skim milk powder at room temperature for 60 min. The samples were then incubated with primary antibody overnight at 4˚C, followed by incubation with the secondary antibody at room temperature for 1 h. Chemiluminescent color development was performed by adding the developing solution.

### Mitochondrial membrane potential assay

Mitochondrial membrane potential was assayed using the JC-1 probe according to the manufacturer's instructions. When the mitochondrial membrane potential was high, JC-1 aggregated in the mitochondrial matrix and formed a polymer, producing red fluorescence; when the mitochondrial membrane potential was low, JC-1 did not aggregate in the mitochondrial matrix, and the monomers produced green fluorescence. Images were obtained using a fluorescence microscope (Olympus, Japan), and the fluorescence intensity was analyzed using

ImageJ. The ratio of red to green fluorescence was used to measure the change in mitochondrial membrane potential.

## Quantitative real-time PCR

Total RNA was extracted using TransZol (Transgen Biotech, Beijing, China), and reverse transcription and qRT-PCR were performed according to the instructions provided with the relevant kits. *GAPDH* was selected as the internal reference gene. The sequences were as follows: UCP2 (F 5′-GCAGTTCTACACCAAGGGCT-3′, R 5′-GGAAGCGGACCTTTACCACA-3′), siUCP2(F 5′-AGAGCACUGUCGAAGC CUACA-3′, R 5′-UAGGCUUCGACAGUGCUCUGG-3′), and GAPDH (F 5′-AGTTCA ACGGCACAGTCAAGGC-3′, R 5′-CGACATACTCAGCACCAGCA TCAC-3′). The relative expression was calculated using the $2^{-\Delta\Delta CT}$ method.

## Oxidative stress and ATP assays

According to the manufacturer's instructions, oxidative stress levels were measured using SOD, GSH-Px, MDA and NADPH kits, and cellular ATP levels were measured using ATP kits. Absorbance values were measured at different wavelengths using an enzyme meter and analyzed according to the standard curves and corresponding formulas.

## ELISA

Cardiomyocyte injury was assessed using ELISA kits for TNNT2, CK-MB, and BNP in rat serum according to the manufacturer's instructions.

## Transmission electron microscopy

Different groups of rat myocardial specimens and different drug-stimulated H9c2 cells were fixed with 2.5% glutaraldehyde phosphate and stained with1% phosphotungstic acid. The mitochondrial ultrastructure was observed and analyzed by using a Transmission Electron Microscope AMT Imaging System (Advanced Microscopy Techniques Co, USA) at magnifications of 5000×, 8000×, and 25000×.

## HE and masson staining

Rat myocardial tissues were fixed with 4% paraformaldehyde, embedded in paraffin, and cut into 3-μm-thick wax slices. The sections were stained with hematoxylin and eosin (HE), Masson Lichtenstein acidic reagent, and toluidine blue and observed under a light microscope (Olympus, Japan).

## TUNEL staining

H9c2 cells were fixed with 4% paraformaldehyde, punched with 0.3% TritonX-100, stained with TUNEL working solution under light-avoidance conditions (37°C for 60min) and stained with DAPI (C1005, Beyotime, China). The cells were then observed under a fluorescence microscope. Rat cardiac muscle tissues were fixed with 4% paraformaldehyde, embedded in paraffin, and then cut into wax slices with a thickness of 3μm. Following this, they were stained with TUNEL and observed under a light microscope.

## Statistical analyses

All statistical analyses were performed using GraphPad Prism 9.0. Datas are expressed as the mean ± standard deviation (SD). Comparisons between two groups were performed using

Student's *t*-test, comparisons among multiple groups were performed using one-way ANOVA followed by Tukey's post hoc test. Statistically different at p<0.05, statistically significant at p<0.01. All data used in statistical analyses were obtained from three or more independent repeated experiments.

## Results

### DOX induces H9c2 cell damage

H9c2 cells were stimulated with various concentrations of DOX (0, 0.5, 1, or 2 μmol/L) for 24 h for CCK-8 detection. Exposure to 1 μmol/L DOX for 24 h decreased H9c2 cell viability by approximately 50% (compared with that in the control group), and the DOX-induced decrease in cell viability was dose-dependent. We stimulated H9c2 cells with 1 μmol/L DOX for different durations (0, 6, 12, 24, and 48 h). A CCK-8 assay showed that cell viability decreased by approximately 50% at 24 h. Therefore, we stimulated H9c2 cells with 1 μmol/L DOX for 24 h for subsequent experiments (Fig 1A and 1B).

Free radical production is the main cause of cardiomyocyte damage by DOX, and cardiotoxicity occurs progressively with ROS production and lipid peroxidation [35]. As determined

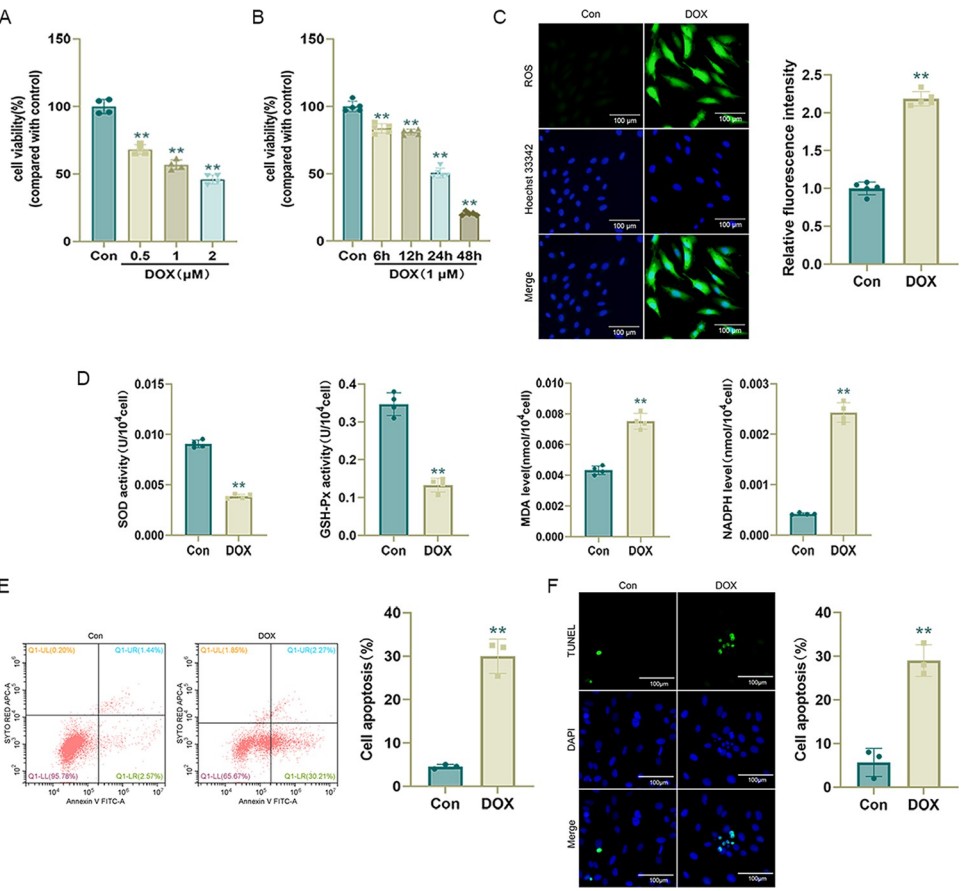

**Fig 1. DOX induces H9c2 cell damage.** (A) and (B) Cell viability determined by CCK-8 assays after treatment with DOX at different concentrations for 24 h and treatment with 1 μmol/L DOX for different times (n = 4 or 5); (C) Representative DCFH-DA images and statistical results (n = 5); (D) SOD, GSH-Px, MDA, and NADPH levels in H9c2 cells (n = 4); (E) Apoptosis rate measured by flow cytometry (n = 3). (F) Representative TUNEL staining images and statistical results (n = 3). Values represent the mean±SD.*p<0.05 vs. Con group,**p<0.01 vs. Con group.

using the DCFH-DA probe, DOX increased ROS levels in cardiomyocytes (Fig 1C), resulting in decreased SOD and GSH-Px activity and increased MDA and NADPH levels (Fig 1D), suggesting that DOX causes oxidative stress injury in cardiomyocytes. Flow cytometry and TUNEL staining revealed that the apoptosis rate was significantly higher ($p < 0.01$) in the DOX group than in the Con group (Fig 1E and 1F), suggesting that DOX caused apoptosis in H9c2 cells.

## DOX induces mitochondrial damage in H9c2 cells

Previous studies have shown that DOX can lead to cardiomyocyte apoptosis via endogenous pathways [36], particularly the mitochondrial pathway. Furthermore, DOX can lead to mitochondrial damage [5]. In this study, DOX increased the expression levels of the apoptosis-related protein Bax, decreased expression levels of Bcl-2, and increased expression levels of Cleaved Caspase-3/Caspase-3 (Fig 2A), indicating that DOX promotes apoptosis in cardiomyocytes and its mechanism of action involves mitochondria. We further evaluated mitochondrial membrane potential and ATP levels, revealing that DOX could cause a decrease in mitochondrial membrane potential and ATP levels in cardiomyocytes (Fig 2B and 2C), while mitochondrial damage (mitochondrial structural disorganization, fragmentation, and cristae rupture) was observed by transmission electron microscopy (Fig 2D).

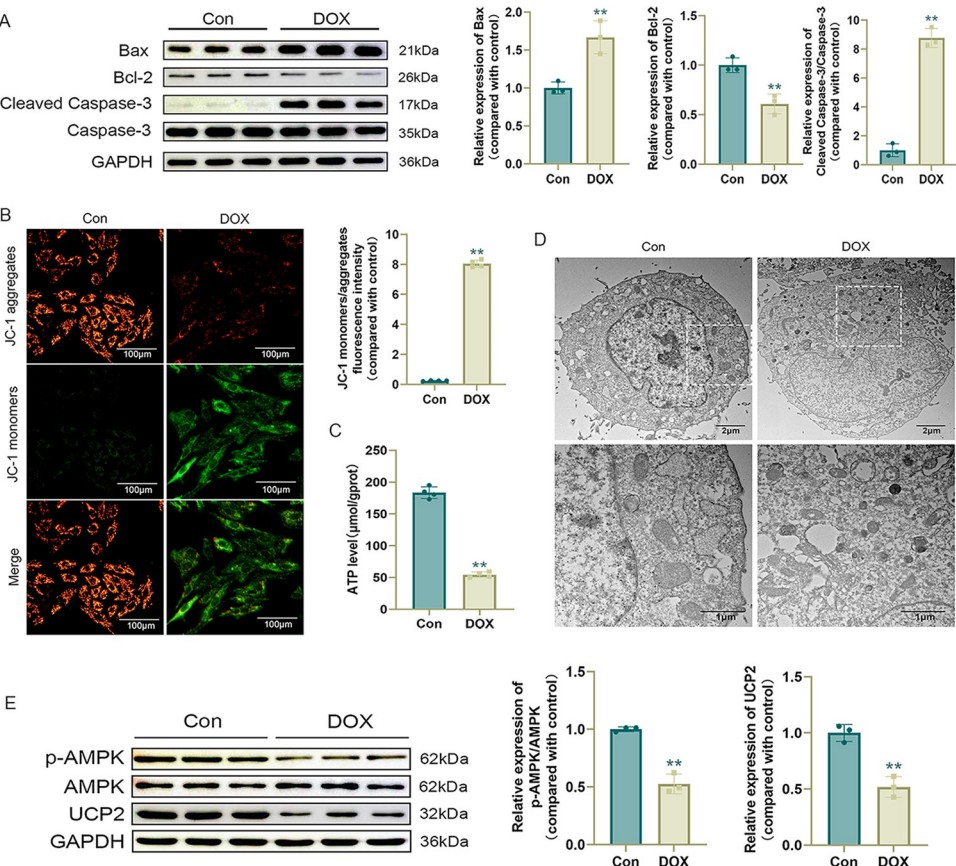

**Fig 2. DOX induces mitochondrial damage in H9c2 cells.** (A) Western blot detection of apoptosis-related protein levels and statistical results (n = 3); (B) Representative JC-1 images and quantification of fluorescence intensity for JC-1 monomers/aggregates (n = 4); (C) ATP level (n = 4); (D) Representative images of mitochondria in H9c2 cells observed by transmission electron microscopy; (E) Western blot detection of p-AMPK, AMPK, and UCP2 levels and statistical results (n = 3). Values are presented as the mean ± SD. *p<0.05 vs. Con group, **p<0.01 vs. Con group.

We performed AMPK and UCP2 assays. Western blotting showed that DOX treatment resulted in decreased levels of p-AMPK/AMPK and UCP2 in cardiomyocytes (Fig 2E), suggesting that the damage to cardiomyocytes caused by DOX may be related to AMPK/UCP2.

## AP39 ameliorates DOX-induced myocardial injury

AP39 has a concentration-dependent effect on mitochondrial activity. At low concentrations (30–100 nmol/L), AP39 stimulates mitochondrial electron transport and cellular bioenergetic functions, and at high concentrations (300 nmol/L), it has an inhibitory effect on mitochondrial activity [19]. Therefore, we first stimulated H9c2 cells with different concentrations of AP39 (0, 30, 50, 100, 300, and 500 nmol/L) for 24 h and performed CCK-8 assays. The results were in accordance with those of previous reports indicating that AP39 at lower concentrations (30–100 nmol/L) does not significantly reduce cell viability. A decrease in cell viability was detected at 300 nmol/L, and a significant decrease in cell viability was detected at 500 nmol/L. Subsequently, we co-stimulated H9c2 cells with 1 µmol/L DOX and different concentrations of AP39 for 24 h. The CCK-8 results showed that the improvement in cell viability was statistically significant at AP39 concentrations of 50 nmol/L and 100 nmol/L, and the improvement was particularly obvious at an AP39 concentration of 100 nmol/L. Owing to this, we chose 100 nmol/L AP39 for subsequent experiments (Fig 3A–3C).

We measured the intracellular $H_2S$ content under different conditions, demonstrating that DOX stimulation decreases $H_2S$ content in H9c2 cells, and this decrease was attenuated by the exogenous administration of AP39 (Fig 3D). In addition, AP39 significantly ameliorated DOX-induced oxidative stress injury in H9c2 cells, with a significant decrease in intracellular ROS levels (Fig 3E), improvements in SOD and GSH-Px activity, and decreases in MDA and NADPH levels after co-treatment with AP39 compared with corresponding levels in the DOX group (Fig 3F). AP39 ameliorated DOX-induced cardiomyocyte apoptosis, which was significantly lower in the DOX+AP39 group than in the DOX group (Fig 3G and 3H).

## AP39 ameliorates DOX-induced mitochondrial damage

We further investigated the mechanisms by which AP39 exerted protective effects against DOX-induced myocardial injury. Western blotting showed that AP39 decreased the expression of the apoptosis-related proteins Bax and Cleaved Caspase-3/Caspase-3 and increased the expression of Bcl-2 (Fig 4A). Additionally, AP39 attenuated the DOX-induced decrease in mitochondrial membrane potential and ATP levels in cardiomyocytes (Fig 4B and 4C). Transmission electron microscopy revealed that mitochondrial damage (i.e., the disorganization of mitochondrial structure, fragmentation, and cristae breakage) was attenuated by AP39 (Fig 4D).

As determined by western blotting, cardiomyocyte p-AMPK/AMPK and UCP2 levels were elevated after co-treatment with AP39 and DOX compared to after DOX stimulation alone (Fig 4E), suggesting that the beneficial effect of AP39 on DOX cardiotoxicity may be related to AMPK/UCP2.

## Inhibition of AMPK expression limits the beneficial effect of AP39 on DOX-induced cardiotoxicity

To verify whether the beneficial effect of AP39 on DOX cardiotoxicity was related to AMPK, we inhibited AMPK using the AMPK inhibitor Compound C (CC) and demonstrated the effectiveness of CC by western blotting (Fig 5A). As determined by a CCK-8 assay, CC did not influence cell viability (Fig 5B). ROS levels were significantly higher in the DOX+AP39+CC group than in the DOX+AP39 group (Fig 5C). SOD and GSH-Px activities were lower and

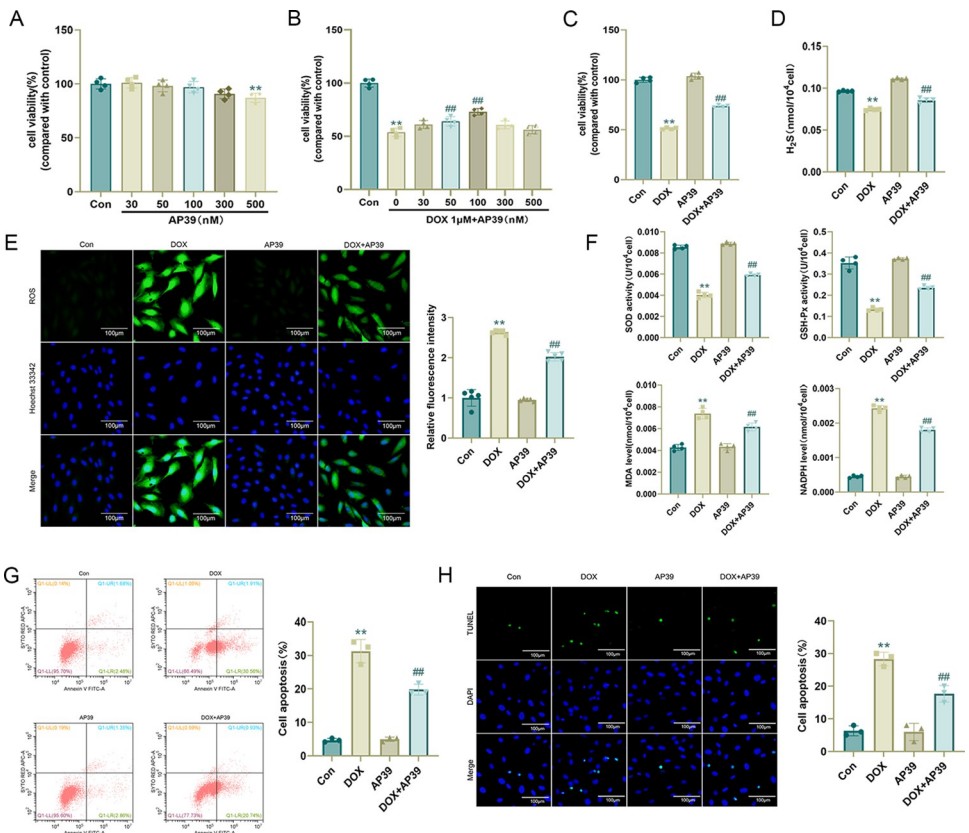

**Fig 3. AP39 ameliorates DOX-induced myocardial injury.** (A)-(C) Cell viability determined by CCK-8 assays after H9c2 cells were treated with different concentrations of AP39 for 24 h, 1 μmol/L DOX and different concentrations of AP39 for 24 h (n = 4); (D) $H_2S$ content in cells of each group (n = 4); (E) Representative DCFH-DA images and statistical results (n = 5); (F) SOD, GSH-Px, MDA, and NADPH levels in H9c2 cells (n = 4); (G) Apoptosis rate measured by flow cytometry (n = 3). (H) Representative TUNEL staining images and statistical results (n = 3). Values are presented as the mean±SD. *p<0.05 vs. Con group, **p<0.01 vs. Con group. #p<0.05 vs. DOX group, ##p<0.01 vs. DOX group.

MDA and NADPH levels were higher in the DOX+AP39+CC group than in the DOX+AP39 group (Fig 5D), suggesting that the inhibition of AMPK expression limited the beneficial effect of AP39 on DOX-induced oxidative stress injury in cardiomyocytes. The apoptosis rate was higher in the DOX+AP39+CC group than in the DOX+AP39 group (Fig 5E and 5F). Western blotting showed that the expression levels of Bax and Cleaved Caspase-3/Caspase-3 were higher and expression levels of Bcl-2 were lower in the DOX+AP39+CC group than in the DOX+AP39 group (Fig 5G), suggesting that the inhibition of AMPK limited the effect of AP39 on DOX-induced apoptosis in cardiomyocytes. Furthermore, the beneficial effects of AP39 on both mitochondrial membrane potential and ATP levels in cardiomyocytes were weakened by the inhibition of AMPK expression (Fig 5H and 5I). These results suggest that AP39 ameliorates DOX-induced cardiotoxicity by regulating AMPK expression.

We further evaluated the regulatory relationship between AMPK and UCP2. Although the down-regulation of UCP2 by DOX was improved by co-treatment with AP39, the expression level of UCP2 in the DOX+AP39+CC group was still significantly lower than that in the DOX+AP39 group (Fig 5J), indicating that the inhibition of AMPK expression suppressed the up-regulation of UCP2 by AP39. These findings suggest that UCP2 may function downstream of AMPK in the regulation of DOX-induced cardiotoxicity by AP39.

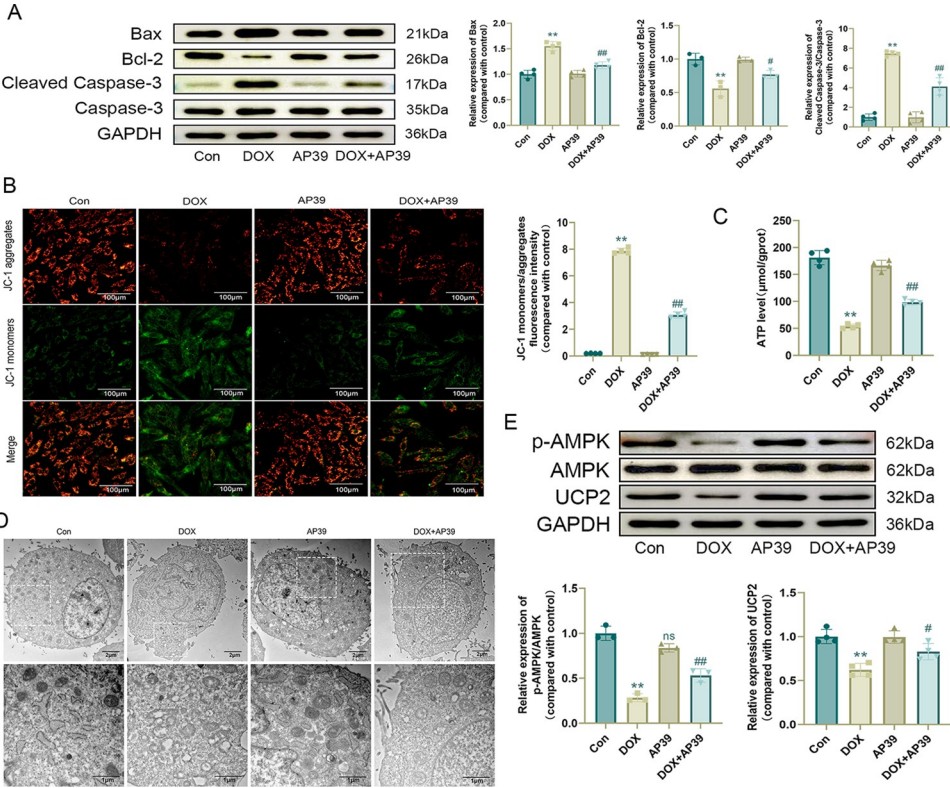

**Fig 4. AP39 ameliorates DOX-induced mitochondrial damage.** (A) Western blot detection of apoptosis-related protein levels and statistical results (n = 3 or 4); (B) Representative JC-1 images and quantification of fluorescence intensity for JC-1 monomers/aggregates (n = 4); (C) ATP level (n = 4); (D) Representative images of mitochondria in H9c2 cells observed by transmission electron microscopy; (E) Western blot detection of p-AMPK, AMPK, and UCP2 levels and statistical results (n = 3 or 4). Values are presented as the mean ± SD. *$p < 0.05$ vs. Con group, **$p < 0.01$ vs. Con group. #$p < 0.05$ vs. DOX group, ##$p < 0.01$ vs. DOX group. NS indicates no significant difference vs. Con group.

## AP39 improves DOX-induced cardiotoxicity by preventing the down-regulation of UCP2

To clarify whether the beneficial effect of AP39 on DOX-induced cardiotoxicity was achieved by modulating the expression of UCP2, we inhibited the expression of UCP2 using small interfering RNA and confirmed the effectiveness of transfection by qRT-PCR and western blotting (Fig 6A). CCK-8 results showed that cell viability did not differ significantly in the NC and siUCP2 groups compared with the Con group (Fig 6B). Oxidative stress damage, apoptosis, and mitochondrial damage were not significantly improved in the DOX+AP39+siUCP2 group compared with those in the DOX+AP39 group. In particular, ROS levels were high (Fig 6C), SOD and GSH-Px levels were low, and MDA and NADPH levels were high (Fig 6D). The apoptosis rate remained high, and western blotting showed that Bax and Cleaved Caspase-3/Caspase-3 levels were high and Bcl-2 levels were low (Fig 6E–6G). Mitochondrial membrane potential and ATP levels remained low (Fig 6H and 6I). The above results suggested that the inhibition of UCP2 inhibited the beneficial effect of AP39 on DOX cardiotoxicity, signifying that UCP2 mediates the effects of AP39.

We hypothesized that UCP2 acts downstream of AMPK. To verify this, we further evaluated the levels of p-AMPK/AMPK. DOX decreased the expression of p-AMPK/AMPK. AP39 upregulated p-AMPK/AMPK, and the inhibition of UCP2 did not influence the effect of AP39

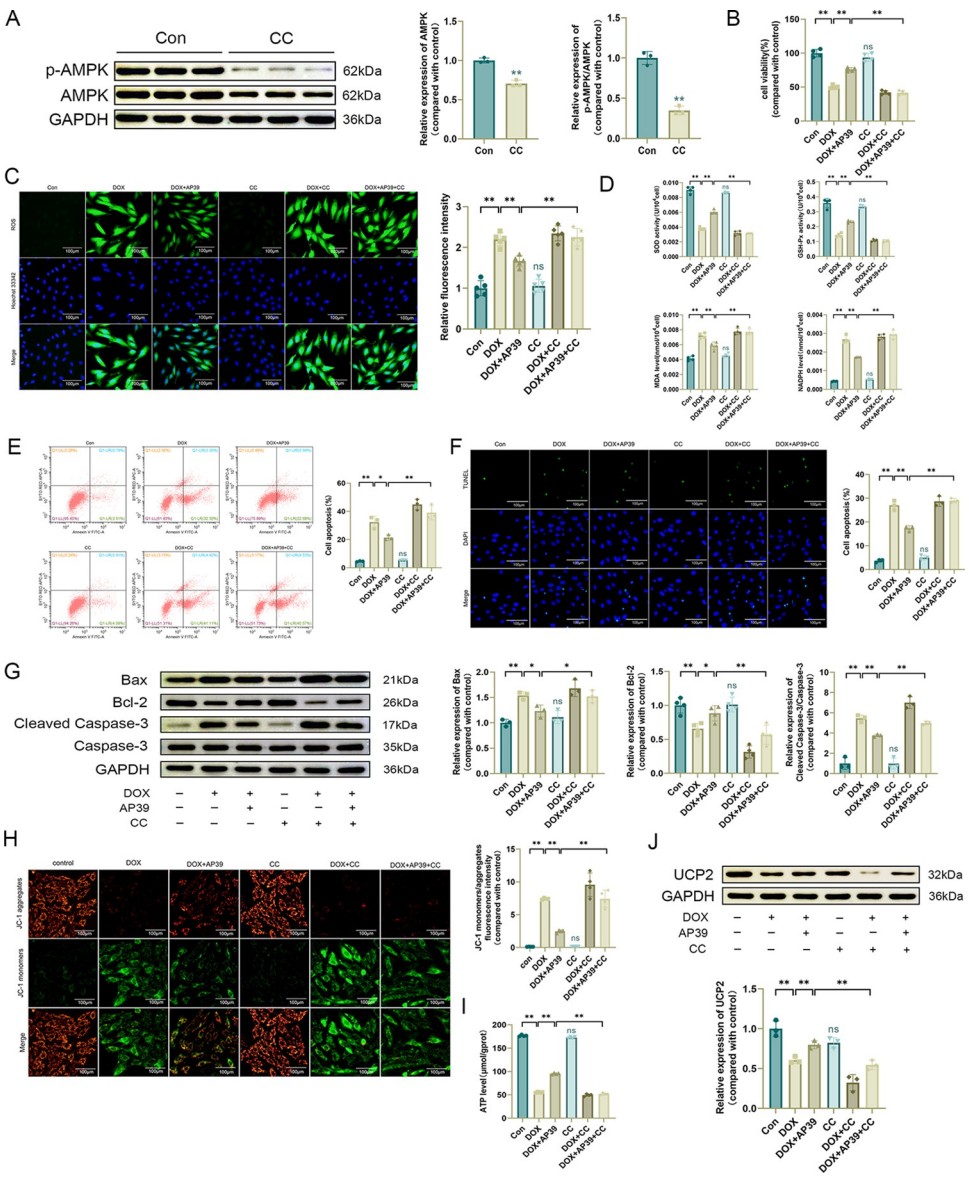

**Fig 5. Inhibition of AMPK expression limits the beneficial effect of AP39 on DOX-induced cardiotoxicity.** (A) Western blot detection of p-AMPK and AMPK levels and statistical results (n = 3); (B) CCK-8 assay of cell viability (n = 4); (C) Representative DCFH-DA images and statistical results (n = 5); (D) SOD, GSH-Px. MDA, and NADPH levels in H9c2 cells (n = 4); (E) Apoptosis rate measured by flow cytometry (n = 3); (F) Representative TUNEL staining images and statistical results (n = 3). (G) Western blot detection of apoptosis-related protein levels and statistical results (n = 3 or 4); (H) Representative JC-1 images and quantification of fluorescence intensity for JC-1 monomers/ aggregates (n = 4); (I) ATP level (n = 4); (J) Western blot detection of UCP2 levels and statistical results (n = 3). Values are presented as the mean±SD.*p<0.05,**p<0.01. NS indicates no significant difference vs. Con group.

([Fig 6J]). We previously confirmed that the inhibition of AMPK could affect the expression of UCP2; therefore, these findings further demonstrated that UCP2 functions downstream of AMPK.

Collectively, these findings demonstrated that AP39 ameliorates DOX-induced oxidative stress damage, apoptosis, and mitochondrial damage in H9c2 cells by regulating the expression of AMPK/UCP2.

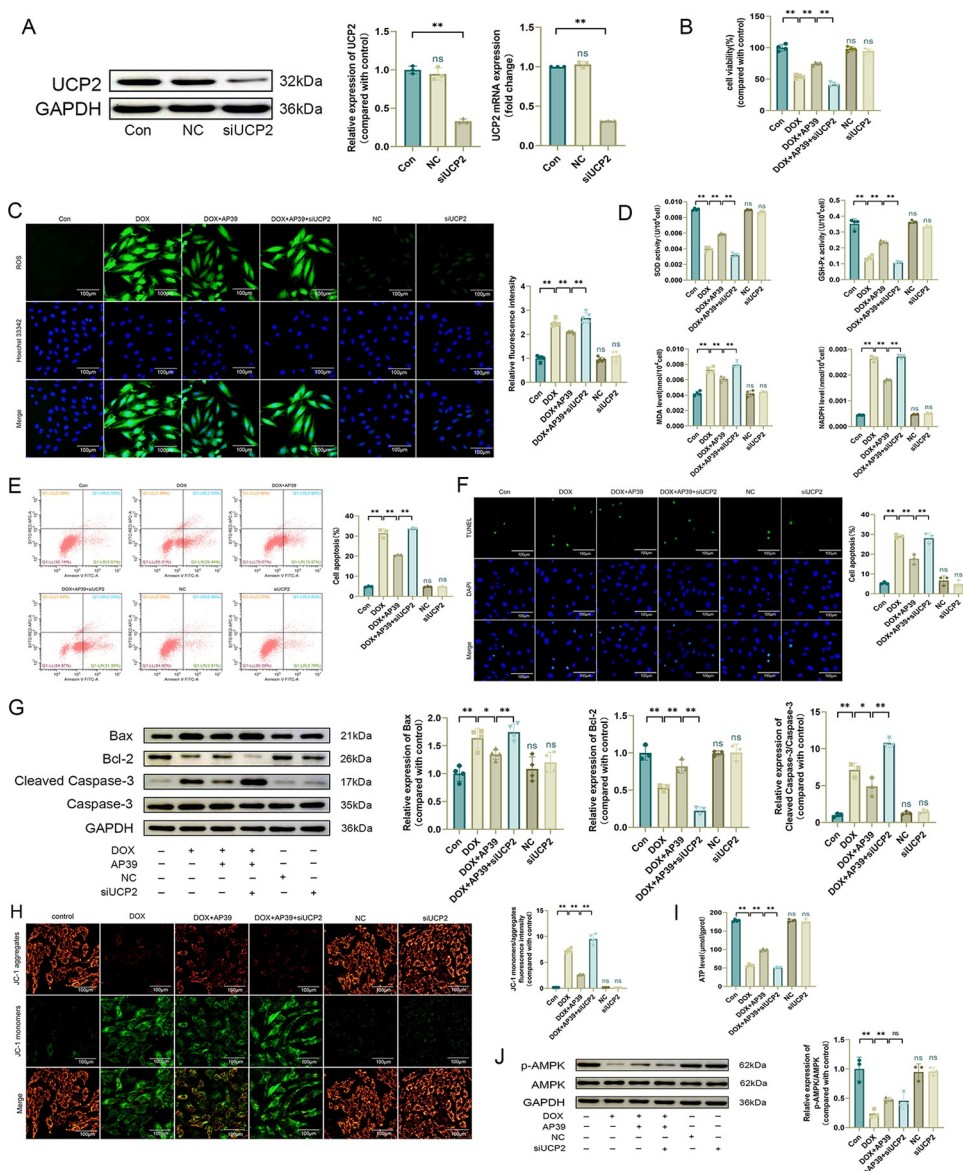

**Fig 6. AP39 improves DOX-induced cardiotoxicity by preventing the down-regulation of UCP2.** (A) Western blot analysis of UCP2 (n = 3) and qRT-PCR for UCP2 mRNA levels (n = 3); (B) CCK-8 assay of cell viability (n = 4); (C) Representative DCFH-DA images and statistical results (n = 5); (D) SOD, GSH-Px, MDA, and NADPH levels in H9c2 cells (n = 4); (E) Apoptosis rate measured by flow cytometry (n = 3); (F) Representative TUNEL staining images and statistical results (n = 3). (G) Western blot detection of apoptosis-related protein levels and statistical results (n = 3 or 4); (H) Representative JC-1 images and quantification offluorescence intensity for JC-1 monomers/aggregates (n = 4); (I) ATP levels (n = 4); (J) Western blot detection of p-AMPK and AMPK levels and statistical results (n = 3). Values are presented as the mean±SD.*p<0.05,**p<0.01. NS indicates no significant difference vs. Con group.

## AP39 attenuates DOX-induced cardiotoxicity in rats by regulating the AMPK/UCP2 pathway

To further validate our experimental results, we conducted in vivo experiments in rats. DOX administration resulted in a significant decrease in body weight and an elevated heart/body weight ratio in rats compared with those in the control group (Fig 7A and 7B). Cardiac ultrasound showed a significant decrease in EF%, FS%, and E/A, suggesting that there was a

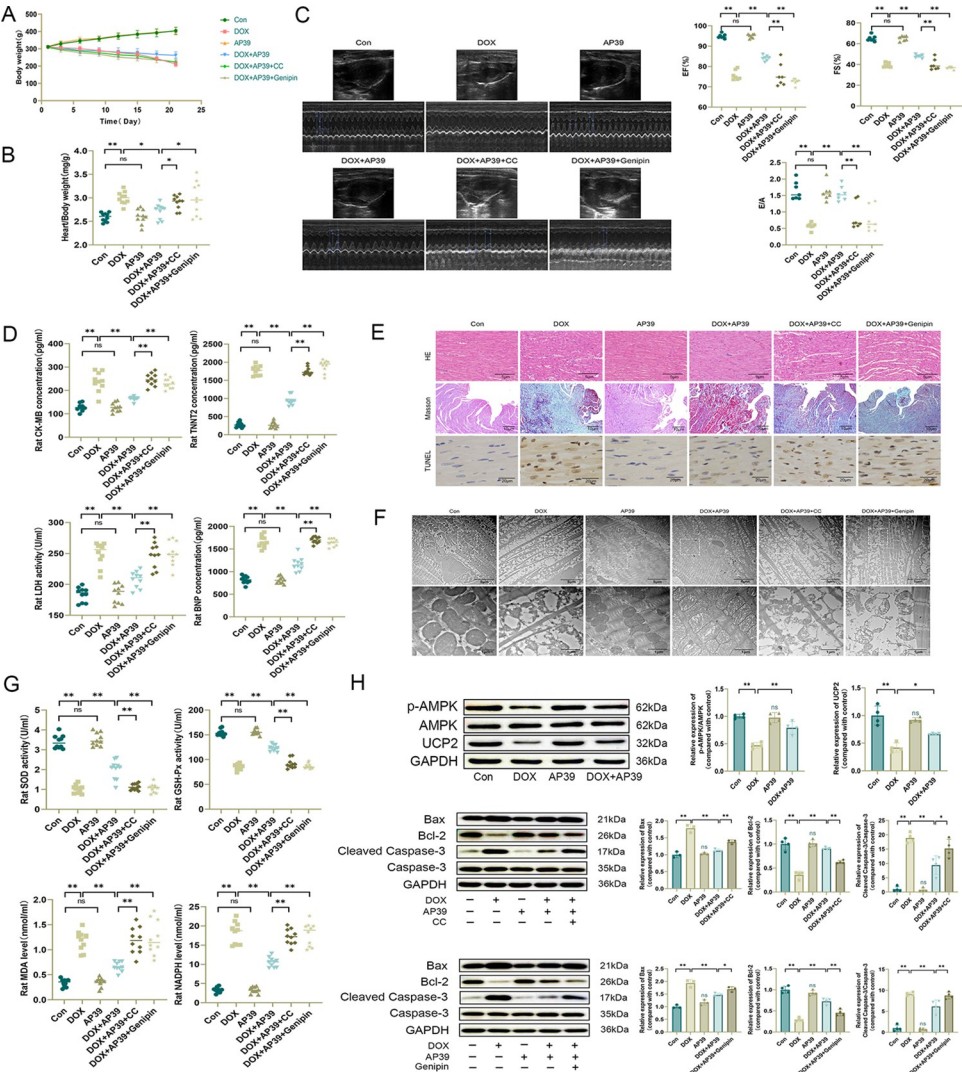

**Fig 7. AP39 ameliorates DOX-induced cardiotoxicity in rats by regulating the AMPK/UCP2 pathway.** (A) Body weight in different groups of rats (n = 10); (B) Heart/body weight ratio in different groups of rats (n = 10); (C) Representative echocardiographic images and quantitative analysis of EF%, FS%, and E/A (n = 7); (D) Serum TNNT2, CK-MB, LDH, and BNP levels in rats (n = 10); (E) Representative HE, Masson and TUNEL staining images of the rat myocardium;(F) Representative images of mitochondria in rat cardiomyocytes observed by transmission electron microscopy; (G) SOD, GSH-Px, MDA, and NADPH levels in rat serum (n = 10); (H) Westen blot detection of apoptosis-related protein, p-AMPK, AMPK and UCP2 levels and statistical results (n = 3 or 4). Values are presented as the mean±SD.*p<0.05,**p<0.01. NS indicates no significant difference vs. Con group.

significant decline in cardiac function (Fig 7C). The levels of TNNT2, CK-MB, LDH, and BNP were significantly increased in abdominal aorta blood after DOX administration (Fig 7D), indicating obvious myocardial damage. HE staining of the rat myocardium was observed under an optical microscope; myocardial cells in the DOX group were deformed, broken, and dissolved, with edema, an enlarged myocardial interstitial space, unevenly colored myocardial fibers, and inflammatory cell infiltration. Masson staining showed a disrupted arrangement of cardiomyocytes, obvious increase in blue collagen fibers in the interstitium of the myocardium, and obvious myocardial fibrosis in the DOX group. A large number of TUNEL positive nuclei were observed in the myocardial tissues of TUNEL staining showed DOX group, with

different shapes and sizes and uneven distribution (Fig 7E). Mitochondrial swelling, structural disorder, fragmentation, ridge breakage, and vacuole-like degeneration of cardiomyocytes in the DOX group were observed by transmission electron microscopy (Fig 7F). We also tested indexes of serum oxidative stress in rats. SOD and GSH-Px activities were lower and MDA and NADPH levels were higher in the DOX group than in the control group (Fig 7G). Western blotting showed that DOX increased the expression of the apoptosis-related protein Bax, decreased the expression of Bcl-2, increased the expression of Cleaved Caspase-3/Caspase-3, and decreased expression levels of p-AMPK/AMPK and UCP2 (Fig 7H). In the DOX+AP39 group, the toxic effects of DOX were ameliorated to varying degrees, consistent with the results of our in vitro experiments. We also confirmed the mechanism by which AP39 improves DOX cardiotoxicity in vivo by administering the AMPK inhibitor CC and UCP2 inhibitor genipin. Cardiomyocyte injury, oxidative stress injury, mitochondrial injury, and apoptosis, as described above, did not differ significantly in the DOX+AP39+CC and DOX+AP39+genipin groups, which was consistent with results of in vitro experiments. These findings suggest that the beneficial effect of AP39 on DOX-induced cardiotoxicity in rats is achieved by modulating AMPK/UCP2 expression.

## Discussion

DOX is a broad-spectrum, highly effective anthracycline-based antitumor drug commonly used to treat different types of tumors. It can significantly improve the survival rate of patients with cancer. However, its severe cardiotoxicity greatly limits its application. It has been shown that anthracyclines are strongly associated with up to 5% left ventricular dysfunction, reduced left ventricular ejection fraction and symptomatic heart failure [37]. The complications of anthracyclines are dose-dependent, with a cumulative drug dose concentration of 400 mg/m$^2$ leading to heart failure in 3.5% of patients, and the incidence of cardiotoxicity ranging from 7–16% at a cumulative dose of 550 mg/m$^2$, and from 18–48% at a dose of 700 mg/m$^2$ [38]. This not only limits the therapeutic dose of DOX, but also greatly affects the quality of the patients and can even shorten their life expectancy. Therefore, there is an urgent need to find drugs that can reduce the cardiotoxicity of DOX. In this study, both in vivo and in vitro experiments demonstrated that the exogenous mitochondria-targeted H$_2$S donor AP39 could attenuate DOX-induced cardiotoxicity by ameliorating oxidative stress, apoptosis, and mitochondrial damage. Mechanistically, we found that AP39 exerts its protective effects by activating the expression of AMPK/UCP2, and inhibitors of AMPK and UCP2 can attenuate or even eliminate the beneficial effect of AP39. These results clearly indicate that AP39 is promising for the prevention or treatment of DOX cardiotoxicity.

Increasing focus on DOX cardiotoxicity has led to extensive research. Studies have shown that DOX decreases levels of SOD, CAT, and GSH-Px and increases levels of MDA in the rat heart, and the amelioration of oxidative stress injury can ameliorate cardiotoxicity [39,40], consistent with our findings. In our experiments, DOX induced ROS production in H9c2 cardiomyocytes, decreased SOD and GSH-Px activity in cardiomyocytes and rat serum, and increased MDA and NADPH levels, indicating that it induces oxidative stress in cardiomyocytes. DOX can induce cardiomyocyte apoptosis through both endogenous and exogenous pathways [36]. For example, DOX can induce apoptosis and pyroptosis via the Akt/mTOR signaling pathway [41], heat shock proteins (HSP-10, HSP-20, HSP-22, HSP-27, and HSP-60), and lipocalin, and it is possible to reduce the cardiotoxicity of DOX by promoting antiapoptotic activity [42,43], as demonstrated herein. In particular, we found that DOX can significantly increase the apoptosis rate of H9c2 cells, up-regulate Bax and Cleaved Caspase-3/Caspase-3, and down-regulate Bcl-2, suggesting that DOX can induce endogenous apoptosis via the

mitochondrial pathway. Compared with other cell types, cardiomyocytes have more mito-chondria, and DOX mainly acts on cardiomyocyte mitochondria, interfering with mitochon-drial electron transport, leading to the formation of superoxide (O2-) free radicals [44]. DOX induces mitochondrial DNA (mtDNA) mutations and defects along with elevating ROS in mitochondria, and these changes have been implicated in the development of cardiomyopathy [45]. DOX can also induce excessive opening of the mitochondrial permeability transition pore [46] and affect mitochondrial KATP channel activity [47], leading to myocardial injury. In our experiments, DOX decreased mitochondrial membrane potential and ATP levels in car-diomyocytes. Mitochondrial structure disorganization, fragmentation, and cristae rupture were observed. These in vivo and in vitro experiments clearly show that DOX causes structural damage and dysfunction of mitochondria in cardiomyocytes. In recent years, researchers have been actively exploring the mechanism of DOX-induced cardiotoxicity and searching for ways to ameliorate it. Abdel-Daim et al. found that DOX could lead to increased levels of pro-inflammatory factors IL-1β, TNF-α and degenerative changes in myocardial tissues of rats, and allicin was able to attenuate the apoptosis and inflammatory response induced by DOX [48]. Metformin can alleviate DOX-induced cardiac injury by increasing PDGFR expression and decreasing $H_2O_2$ levels, activating the AMPK signaling pathway [27,49]. Sildenafil, a PDE-5 inhibitor, demonstrated a protective effect in Dox-induced cardiotoxicity by modulating the NO/cyclic GMP, mitochondrial $K^+$ATP channel and oxidative stress [50]. In addition, phyto-chemicals also exhibit cardioprotective effects by exerting antioxidant, anti-inflammatory and anti-apoptotic activities, as well as regulating lipid metabolism and intracellular calcium homeostasis [6].

Hydrogen sulfide ($H_2S$), initially described as a toxic gas with a rotten egg odor, is similar in nature to nitric oxide (NO) and carbon monoxide (CO), endogenous gaseous signaling molecules in mammals. Increasing studies have shown that it is involved in a variety of patho-physiological processes, such as oxidative stress, inflammation, apoptosis, and angiogenesis; additionally, it plays a protective role in the pathogenesis and progression of cardiovascular diseases [51]. $H_2S$ reduces lipid peroxidation by hydrogen peroxide and superoxide scavenging in a model of isoprenaline-induced myocardial injury [52]. $H_2S$-mediated activation of Nrf2-dependent pathways leads to the upregulation of genes involved in endogenous antioxi-dant defense [53]. It protects mitochondrial function by inhibiting respiration, thereby limit-ing ROS production and reducing mitochondrial uncoupling [54]. Furthermore, $H_2S$ significantly prevents high glucose-induced apoptosis in cardiomyocytes by modulating the expression of Bax and Bcl-2 [55]. AP39, a novel mitochondria-targeted $H_2S$ donor, can amelio-rate high-fat-diet-induced liver injury in young rats by attenuating oxidative stress and mito-chondrial damage [56]. It can support cellular bioenergetics and prevent Alzheimer's disease by maintaining mitochondrial function in APP/PS1 mice and neurons [31]. It can prevent 6-hydroxydopamine-induced mitochondrial dysfunction [57]. In this study, both in vivo and in vitro experiments confirmed that exogenous mitochondrial targeting of AP39 ameliorates DOX-induced oxidative stress by decreasing cardiomyocyte ROS levels, elevating SOD and GSH-Px contents, and decreasing MDA and NADPH levels; it improved cardiomyocyte apo-ptosis by regulating the expression of apoptosis-related proteins, such as Bax, Bcl-2, and Cleaved Caspase-3/ Caspase-3, and improved DOX-induced mitochondrial injury by elevating mitochondrial membrane potential and ATP levels, consistent with results of previous studies on the mechanisms underlying the myocardial protective effects of $H_2S$ or AP39.

Cardiac tissues have high metabolic energy requirements, and growing evidence suggests that AMPK plays a key role as an energy sensor and a major regulator of metabolism in regu-lating cell survival in vivo and in vitro [58]. In 2005, Tokarska-Schlattner et al. were the first to demonstrate that AMPK inactivation plays an important role in DOX cardiotoxicity [59].

Since then, additional studies have shown that AMPK is closely related to multiple molecular mechanisms underlying DOX-induced cardiomyocyte injury. DOX is able to inhibit the expression and phosphorylation of AMPK proteins in the rat heart via DNA damage-induced Akt signaling, which activates a negative feedback loop of mTOR signaling and leads to cardiac remodeling [60]. DOX can lead to myocardial fibrosis and cardiomyocyte apoptosis in APN-SE mice by inhibiting AMPK expression [61]. Some AMPK activators, such as metformin, statins, resveratrol, and thiazolidinediones, have the potential to prevent DOX cardiotoxicity [62]. Located within the mitochondrial membrane, UCP2 acts as an anion carrier and regulates the transmembrane proton electrochemical gradient in many human tissues; it is involved in a number of processes, including mitochondrial membrane potential, ROS production within the mitochondrial membrane, and calcium homeostasis [63]. UCP2 is involved in the reduction of ROS production and mitochondrial ROS scavenging [64] and can protect cardiomyocytes from oxidative stress by inhibiting ROS production [65]. UCP2 prevents neuronal apoptosis and attenuates brain dysfunction after stroke and traumatic brain injury [66]. It protects the heart from I/R injury by inducing mitochondrial autophagy [67]. Studies on the interaction between AMPK and UCP2 have yielded conflicting results. It has been suggested that UCP2 affects the autophagic process in septic cardiomyopathy via AMPK signaling [68] an regulates cholangiocarcinoma cell plasticity via mitochondrial-AMPK signaling [69]. However, there is substantial evidence that AMPK functions upstream of UCP2. For example, in a model of nonalcoholic fatty liver disease, LB100 regulated UCP2 expression by inhibiting AMPK [34]. Malvidin alleviates mitochondrial dysfunction and ROS accumulation by activating the AMPK-α/UCP2 axis, thereby preventing inflammation and apoptosis in mice with sepsis-associated encephalopathy [26]. Indole sulfate induces oxidative stress and hypertrophy in cardiomyocytes by inhibiting the AMPK/UCP2 signaling pathway [70]. In our experiments, we found that the protective effect of AP39 against DOX cardiotoxicity was mediated by AMPK/UCP2, and the use of AMPK inhibitors affected the expression of UCP2, while the inhibition of UCP2 expression did not have a significant effect on the expression level of AMPK. These findings suggest that AMPK is an upstream signal of UCP2 and regulates the expression of UCP2. The differences in the regulatory relationship between AMPK and UCP2 among studies may be related to differences in disease models, stimuli, and other factors.

Although we confirmed through in vivo and in vitro experiments that AP39 can ameliorate DOX-induced cardiotoxicity by ameliorating oxidative stress, mitochondrial damage, and decreasing apoptosis, and demonstrated that the mechanism of action is related to the modulation of the AMPK/UCP2 pathway by AP39, there are still some limitations to our experiments, which are mainly reflected in the following:(1) The present study clarifies the role of AMPK regulation of UCP2 in the attenuation of DOX cardiotoxicity by AP39; however, the specific mechanism underlying these regulatory effects is not clear. A downstream pathway of AMPK is Sirt1/PGC-1α, and AMPK activates the NAD+-dependent type III deacetylase Sirt1 by increasing the intracellular NAD+/NADH ratio; Sirt1 activation leads to peroxisome proliferation-activated receptor-γ coactivator 1α (PGC-1α) deacetylation and activity regulation, and Sirt1/PGC-1α may be involved in the regulation of UCP2 [24]. Accordingly, the roles of Sirt1/PGC-1α need to be studied further.(2) During our literature review on establishing an animal model of DOX-induced cardiac toxicity, we could not find a relatively uniform standard dose, frequency and cumulative DOX dose. To this end, the dose we used was a comprehensive consideration of various protocols and that used in clinical settings. AP39 is a recently studied mitochondrial targeting H$_2$S donor, and its application in animal experiments has not been widely reported compared to that of inorganic sulfides such as NaHS; therefore, the dose of AP39 and associated experimental results might be closely related to animal species, age, weight, and experimental environment, and changes in any of these factors might have an

impact on experimental findings. (3) Both the in vivo experimental model constructed using Sprague–Dawley rats and the in vitro experimental model of H9c2 cells which simulated cardiomyocytes cannot fully reflect the real pathophysiological changes in the human body, which is a common problem in all experimental studies. We hope that through the continuous development of advanced science and technology such as gene editing, tissue engineering, three-dimensional cell culture, and organoid culture technologies, combined with advancements in bioinformatics, structural biology, biochemistry, and other fields, experimental models will be better optimized to provide a more realistic and effective experimental platform for human disease research. Despite these limitations, our experimental results provide possible therapeutic strategies for DOX cardiotoxicity and support the beneficial effects of AP39. In the future, more in-depth studies can be carried out based on our experimental results to elucidate the specific mechanism of AP39 in improving DOX-induced cardiotoxicity, and discover associated molecular targets, thus providing reliable research to support its use in clinical settings. The findings of this study present the mitochondrial targeted $H_2S$ donor AP39 as a potential therapeutic agent that can improve DOX-induced cardiotoxicity, and possibly having clinical application in treating other cardiovascular diseases.

## Conclusions

Taken together, our findings suggest that AP39 ameliorates DOX cardiotoxicity by attenuating oxidative stress, apoptosis, and mitochondrial damage via modulating the expression of AMPK/UCP2. These findings indicate that AP39 is a promising new therapeutic agent for preventing DOX-induced cardiotoxicity.

## Supporting information

**S1 Raw images. Western blots.**
(PDF)

**S1 Raw data.**
(PDF)

**S1 File. Editing certificate.**
(PDF)

## Author Contributions

**Conceptualization:** Bin Zhang.

**Formal analysis:** Bin Zhang, Yangxue Li.

**Funding acquisition:** Bin Liu.

**Methodology:** Bin Zhang.

**Project administration:** Ning Liu, Bin Liu.

**Resources:** Ning Liu, Bin Liu.

**Supervision:** Bin Liu.

**Validation:** Bin Zhang.

**Writing – original draft:** Bin Zhang.

**Writing – review & editing:** Yangxue Li, Ning Liu.

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
