## [Decision Letter · Decision Letter 0]

9 Jan 2024

PONE-D-23-34322AP39 ameliorates doxorubicin-induced cardiotoxicity by regulating the AMPK/UCP2 pathwayPLOS ONE

Dear Dr. Liu,

Thank you for submitting your manuscript to PLOS ONE. After careful consideration, we feel that it has merit but does not fully meet PLOS ONE’s publication criteria as it currently stands. Therefore, we invite you to submit a revised version of the manuscript that addresses the points raised during the review process.

We look forward to receiving your revised manuscript.

Kind regards,

Mohamed Abdel-Daim, Ph.D.

Academic Editor

PLOS ONE

Journal Requirements:

"This work was supported by Jilin Province Science and Technology Department (20220303002SF), Jilin Provincial Development and Reform Commission (2022C003), Jilin Province Science and Technology Department (20190905002SF)."

Reviewers' comments:

Reviewer's Responses to Questions

**Comments to the Author**

1. Is the manuscript technically sound, and do the data support the conclusions?

Reviewer #1: Yes

Reviewer #2: Partly

2. Has the statistical analysis been performed appropriately and rigorously? 

Reviewer #1: Yes

Reviewer #2: Yes

3. Have the authors made all data underlying the findings in their manuscript fully available?

Reviewer #1: Yes

Reviewer #2: Yes

4. Is the manuscript presented in an intelligible fashion and written in standard English?

Reviewer #1: Yes

Reviewer #2: Yes

5. Review Comments to the Author

Reviewer #1: This study focused on AP39 ameliorates doxorubicin-induced cardiotoxicity by regulating the AMPK/UCP Pathway. The authors found that both in vivo and in vitro experiments showed that DOX induces oxidative stress injury, apoptosis, and mitochondrial damage in cardiomyocytes and decreases the expression of p-AMPK/AMPK and UCP Pathway. AMPK/AMPK and UCP2.All DOX-induced changes were attenuated by AP39 treatment. Furthermore, the protective effect of AP39 was significantly attenuated by the inhibition of AP39. All DOX-induced changes were attenuated by AP39 treatment. Furthermore, the protective effect of AP39 was significantly attenuated by the inhibition of AMPK and UCP2. AMPK/UCP2. This observation is interesting，but there are some questions as follows:

1. caspase-3 and cleaved caspase-3 are shown in all of the authors' figures, but the results are unusual. Normally, if cleaved caspase-3 is increased, then capspase-3 will be decreased, but in the author's results, caspase-3 is not decreased, and many bands are unchanged or even increased. Could the authors explain why?

2. Fig. 2B Why is there a big difference in cell size between Con and Dox groups?

3. the anatomical position of Fig. 7E is obviously different, please use pictures with the same anatomical position.

4. The author's experimental method is relatively limited, only flow cytometry was used to detect apoptosis, which is not sufficient, it is recommended to use TUNEL or DNA laddering to verify their findings.

5. Fig6C why the last two images have small cell size than the other images?

Major:

Figure 7A shows that the body weight of rats in DOX, DOX+AP39+CC and DOX+AP39+Genipin groups was significantly reduced by more than 25% compared with that of normal rats, suggesting that the rats were in a malignant state, which violated animal ethics. Routinely, if the body weight of an animal model decreases by 20%-25%, it suggests that the animal is in a malignant state and has reached the humane endpoint or experimental terminative indicator, requiring euthanasia, rather than continuation of the experiment. Please check the animal ethics requirements of this journal for compliance.

Reviewer #2: 

Experimental Model and Dose Selection: Please further explain the choice of H9c2 cells and Sprague-Dawley rats as experimental models and discuss the limitations of these models in simulating human pathophysiology. Additionally, explain the rationale behind the selected doses and treatment durations for DOX and AP39, ensuring they are relevant to clinical situations.

Data Analysis: Detail the statistical analysis methods used and justify the choice of these methods. Ensure that significance levels and the interpretation of results are accurate and error-free.

Consistency and Reproducibility of Results: Provide additional information about the replicability of the experiments to ensure the reliability and consistency of the results. Describe the details of the control group setup to aid other researchers in replicating the experiment.

Study Limitations: Discuss the limitations of the study, including factors that might affect the conclusions.

Logical Structure: Ensure the paper's clarity and logical flow from the introduction through methods, results, and discussion sections. Provide additional background information on DOX-induced cardiotoxicity to help readers understand its significance and the need for treatment approaches. Enhance the interpretation of experimental results and their relevance. In the discussion, provide future research directions and prospects for potential clinical applications.

Citations and Background Research: Update the references to include the latest and relevant studies to support the research hypothesis and conclusions. Clearly state the motivation and research questions behind the study, and provide background research to illustrate the importance and relevance of the study.

Ethical Compliance: Given the involvement of animal experiments, ensure compliance with all relevant ethical guidelines and clearly state the ethical approval number in the paper.

6. PLOS authors have the option to publish the peer review history of their article (what does this mean?). If published, this will include your full peer review and any attached files.

Reviewer #1: No

Reviewer #2: **Yes: **xiong wei

---

## [Author Response · Author response to Decision Letter 0]

29 Jan 2024

Dear Reviewers: 

Thank you for your review and for the comments regarding our manuscript entitled “AP39 ameliorates doxorubicin-induced cardiotoxicity by regulating the AMPK/UCP2 pathway” (ID: PONE-D-23-34322). The comments were all valuable and very helpful in revising and improving our paper, as well as improving our research. We have addressed all the comments and have made corrections to the manuscript (highlighted in yellow).We hope that the revised manuscript now meets the requirements of the journal. Our point-by-point responses to all comments are presented below:

Reviewer #1:

1.Caspase-3 and cleaved caspase-3 are shown in all of the authors' figures, but the results are unusual. Normally, if cleaved caspase-3 is increased, then capspase-3 will be decreased, but in the author's results, caspase-3 is not decreased, and many bands are unchanged or even increased. Could the authors explain why?

Response: 

Thanks very much for your comment. 

This is a very interesting phenomonon that we noticed during the experiment.We repeated several independent experiments and the results were consistent, just as we presented in the manuscript.

As you mentioned, cleaved caspase-3 is derived from caspase-3, and generally speaking, if cleaved caspase-3 increases, then caspase-3 should decrease.For this purpose, we reviewed the literature and found that other studies have reported results similar to ours.Zhang et al. reported that FNDC5 attenuated DOX-induced cardiomyocyte apoptosis by activating AKT, and verified the expressions of caspase-3 and cleaved caspase-3 in their study. The expression of caspase-3 did not decrease due to the increase of cleaved caspase-3, which was a similar occurrence to our experimental results[1].In another study by Shabaan et al., the authors found that in DOX-induced apoptosis in Wistar rat cardiomyocytes, immunohistochemical staining of rat left ventricular tissues showed a significant elevation in caspase-3 expression [2].Further,Wang et al. found that ropivacaine promoted apoptosis by activating caspase-3 in Bel 7402 and HLE hepatocellular carcinoma cell lines, and although only the expression of cleaved caspase-3 was quantitatively analyzed in their study, both caspase-3 and cleaved caspase-3 were elevated in the reported WB bands [3].

Based on these findings and our results, we may speculate that in our experimental model, the total amount of caspase-3 was not constant but increased, and the expression of cleaved caspase-3 that was sheared off after activation is also increased, so that there will be an increase in cleaved caspase-3 in the WB bands at the same time as a constant or even a slight increase in caspase-3.However, some studies have reported a similar occurrence to what you have proposed, including a study by Wu et al [4], in which caspase-3 expression was decreased when cleaved caspase-3 expression was increased, but the animal species, experimental model, and method of drug administration in their experiments all differed from ours, and these may be factors that led to the contradicting observations.Taken together, we believe that it would be more appropriate to assess the level of apoptosis by using the ratio of cleaved caspase-3/caspase-3.

2. Fig. 2B Why is there a big difference in cell size between Con and Dox groups?

Response: 

Thank you very much for your comment.

In Fig 2B, a JC-1 fluorescent probe was used to detect the mitochondrial membrane potential. When the mitochondrial membrane potential was high, JC-1 aggregated in the mitochondrial matrix and presented red fluorescence. When the mitochondrial membrane potential was low, JC-1 existed as monomers and presented green fluorescence.We considered that the reason for the difference in cell size images between the Con group and DOX groups might be related to the difference in fluorescence intensity. We noticed this occurrence during the study,when we took pictures of cells in different groups under fluorescence microscope, our microscope magnification and fluorescence excitation time were consistent; however, the stronger the fluorescence intensity, the bigger the cells looked and vice versa. This was also the case in Fig 6C as you asked in Question 5. We have attached a picture of JC-1 detection with nucleus staining below as an illustration.Since the JC-1 results mainly compare the intensity of red-green fluorescence, the results we present in the paper do not have merged nuclei.

3. the anatomical position of Fig. 7E is obviously different, please use pictures with the same anatomical position.

Response: 

Thank you very much for your comment.

We have rephotographed the tissue samples under the microscope and revised some of the images in Fig 7E. 

4. The author's experimental method is relatively limited, only flow cytometry was used to detect apoptosis, which is not sufficient, it is recommended to use TUNEL or DNA laddering to verify their findings.

Response: 

Thank you very much for your comment.

We performed western blotting to determine Bax,Bcl-2,caspase-3 and cleaved caspase-3 levels and flow cytometry to determine the apoptosis.We also supplemented both in vivo and in vitro experiments with TUNEL to detect apoptosis, as described in Fig 1F,Fig 3H,Fig 5F,Fig 6F and Fig 7E.

5. Fig6C why the last two images have small cell size than the other images?

Response:

Thank you very much for your comment.

Fig 6C shows the ROS levels in H9c2 cells detected by DCFH-DA fluorescent probe. The higher the intracellular ROS levels, the stronger the green fluorescence intensity.The magnification and fluorescence exposure time of the six images in this group were consistent. The reason why the cell sizes of the last two images are small is because of the different fluorescence intensity between different groups. The stronger the fluorescence intensity, the larger the cells looked and vice versa. Following your question, we realized that the existing images could confuse readers; therefore, we re-tested the ROS levels and performed nucleus staining at the same time, thus making it a clearer comparison between different groups. The results are depicted in Fig 1C,Fig 3E,Fig 5C and Fig 6C.

Major:

Figure 7A shows that the body weight of rats in DOX, DOX+AP39+CC and DOX+AP39+Genipin groups was significantly reduced by more than 25% compared with that of normal rats, suggesting that the rats were in a malignant state, which violated animal ethics. Routinely, if the body weight of an animal model decreases by 20%-25%, it suggests that the animal is in a malignant state and has reached the humane endpoint or experimental terminative indicator, requiring euthanasia, rather than continuation of the experiment. Please check the animal ethics requirements of this journal for compliance.

Response: 

Thank you very much for your comment.

Animal ethics are crucial for conducting animal experiments,and our animal study was approved and supervised by the Animal Protection and Utilization Institutional Committee of Jilin University (Animal Ethics Approval number:2023 No. 463).The end points of our study based on the Animal ethics approval regulations were as follows:if there was a significant weight loss in rats, and the decrease exceeded 25%, DOX was to be stopped to avoid further damage from drug dose accumulation, and the general status of rats was to be carefully evaluated; If there was an obvious state of pain, near-death, or serious adverse damage, experimental euthanasia was to be immediately performed; If the general state of the rats was acceptable, the state of the animal was be continuously observed without the administration of DOX. 

In our study, we noticed a reduction in animal body weight of nearly 25% on the 18th day. However, the general state of the rats was good in three groups. The rats did not appear to be in an obvious state of near-death or obvious pain, the intake of food and water was reduced but could freely be obtained at a normal frequency, and the activity level was reduced but was still good compared with that of rats in the Con group at the same time period. At this point, we had completed all DOX dosing;therefore,we continued to observe the rats for the remaining 3 days.

We love and respect experimental animals.We believe that they have the same right to live and they also experience emotions such as joy and sorrow, pain, and fear. During the study, we treated all animals with kindness, comfort, and minimized their stress, anxiety, and pain,and the whole process is supervised by the Animal Protection and Utilization Institutional Committee of Jilin University.

[References]

1、Zhang X,Hu C,Kong CY,et al.FNDC5 alleviates oxidative stress and cardiomyocyte apoptosis in doxorubicin-induced cardiotoxicity via activating AKT.Cell Death & Differentiation (2020) 27:540–555.doi:org/10.1038/s41418- 019-0372-z.

2、Shabaana DA, Mostafab N, El-Desokyb MM, and Arafat EA.Coenzyme Q10 protects against doxorubicin-induced cardiomyopathy via antioxidant and anti-apoptotic pathway.TISSUE BARRIERS. 2023, VOL. 11, NO. 1, e2019504 (14 pages). doi:10.1080/ 21688370.2021.2019504.

3、Wang WT, Zhu MY, Xu ZX,et al.Ropivacaine promotes apoptosis of hepatocellular carcinoma cells through damaging mitochondria and activating caspase-3 activity.Biol Res. 2019 Jul 12;52(1):36. doi:10.1186/s40659-019 -0242-7.

4、Wu XT, Wang LJ,Wang K,et al.ADAR2 increases in exercised heart and protects against myocardial infarction and doxorubicin-induced cardiotoxicity. Mol Ther. 2022 Jan 5; 30(1): 400–414.doi: 10.1016/j.ymthe. 2021.07.004.

Reviewer #2: 

1.Experimental Model and Dose Selection: Please further explain the choice of H9c2 cells and Sprague-Dawley rats as experimental models and discuss the limitations of these models in simulating human pathophysiology. Additionally, explain the rationale behind the selected doses and treatment durations for DOX and AP39, ensuring they are relevant to clinical situations.

Response:

Thanks very much for your comment.

(1)Reasons and limitations of choosing H9c2 cells and Sprague-Dawley rats as experimental models

In our opinion, it is reasonable to select H9c2 cells and Sprague-Dawley rats as experimental models.H9c2 rat cardiomyocyte is a subclone of the original cloned cell line derived from embryonic BD1X rat heart tissue. It is an in vitro cell model that can be used in place of cardiomyocytes, and can simulate the physiological characteristics of cardiomyocytes.The Sprague-Dawley rat is a common experimental animal model, and its heart structure and function are similar to that of humans. However, these models still have some limitations in simulating human pathophysiology.For example, H9c2 cells cannot fully reproduce the complexity of human cardiomyocytes in vivo under in vitro culture conditions, and lack the regulation of the nervous and endocrine systems. Further,the cells may lose some original biological characteristics after repeated passage. Although Sprague-Dawley rats can simulate the physiological function of the human heart, there are still differences between Sprague-Dawley rats and humans in genetic background, physiological response, metabolism and other aspects, and it is difficult to overcome the influences of species differences. The above limitations are included in our Discussion section.

(2)Dosing and timing of DOX and AP39

The choice of dose and timing of DOX:

In vitro: We used DOX at different concentrations (0.5, 1, 2μM) to stimulate H9c2 cells for 24h, and then conducted CCK-8 cell activity detection.DOX stimulated cell activity was dose-dependent; when DOX concentration was 1μM, H9c2 cells were stimulated for 24h, and cell activity decreased to about 50%.We then used 1μM DOX to stimulate H9c2 cells for different times (6h, 12h, 24h, 48h), and found that the cell activity gradually decreased with the extension of the stimulation time, and after 24h of stimulation, the cell activity decreased to about 50%. Therefore, we selected the experimental condition of stimulating H9c2 cells with 1μM DOX for 24h.These results are reflected in Fig 1A and 1B.

In vivo: There are different literature reports on DOX dosing, and it has been reported that the cumulative dose of DOX will not induce heart failure in rats when it is less than 10mg/kg [1].According to current reports, the most commonly used administration modes in animal experimental models of DOX-induced cardiotoxicity are: DOX 5mg/kg every three days, intraperitoneally (ip),total 20 mg/kg; DOX 6mg/kg once every other day, ip, total 18mg/kg; DOX 5mg/kg/w, ip, total 15mg/kg; DOX 20mg/kg/ day for 5 consecutive days, total volume 100mg/kg; DOX 4mg/kg/w, ip, total 16mg/kg; DOX 2.5mg/kg/w, ip, total 15mg/kg.There is no uniform dosing and frequency of administration.In clinical treatment, patients receiving chemotherapy generally use DOX for several consecutive times in a few months, with a therapeutic dose of 50-75mg/m2 each time, and the cumulative maximum therapeutic dose is about 450mg/m2, which is equivalent to 12mg/kg[2]. Therefore, summarizing the above, a dose of 5mg/kg/w, three times, accumulating to 15mg/kg was chosen for this experiment.

The choice of dosage and timing of AP39:

In vitro: According to relevant reports, AP39 exerts a protective effect on cells at a lower concentration (30 and 100 nM), but at a higher concentration (300 nM), the protective effect may become damaging, and a moderate concentration of AP39(100 nM) can reduce intracellular oxidative stress and maintain cell viability and mitochondrial DNA integrity [3].Therefore, we stimulated H9c2 cardiomyocytes with different concentrations of AP39 (30,50,100,300,500nM) for 24h, and stimulated H9c2 cells with different concentrations of AP39 and 1μM DOX for 24h. The CCK-8 kit was used to detect cell activity. The results showed that, AP39 at 100nM could ameliorate the DOX-induced decline in H9c2 cell activity to the greatest extent, as shown in Fig 3A-C. Therefore, 100 nM of AP39 was selected as the experimental concentration. 

In vivo: According to existing reports, AP39 dosing in animal experiments are as follows:0.1mg/kg/d, ip, for 7 weeks; 50nmol/kg/d, ip, for 4 weeks; 100nmol/kg/d,ip, for 6 weeks. It can be seen that there is no relatively uniform dose and frequency of administration. Since AP39 is not used in clinical application, a clinical dosing reference is not available. Therefore, in our pre-experiments, we first tried AP39 at a dose of 50nmol/kg/d, ip,for 4 weeks, but the rats in the DOX+AP39 group continuously died within 2 weeks, and some of the rats in AP39 group also died. Although the remaining rats were in a normal state, their cardiac function was decreased based on echocardiography. We then tried AP39 at a dose of 50nmol/kg, once every other day, ip, for 3 weeks and AP39 25nmol/kg, once every other day, ip, for 3 weeks. Echocardiography results showed the latter dose had no toxic effect; however, it did not improve cardiac function. With the AP39 50 nmol/kg regimen, rats in the AP39 group had no obvious cardiac function damage, and those in the DOX+AP39 group had significantly improved cardiac function compared with those in the DOX group. Owing to these findings, we chose this as the dosing regimen for experimentation.

2.Data Analysis: Detail the statistical analysis methods used and justify the choice of these methods. Ensure that significance levels and the interpretation of results are accurate and error-free.

Response:

Thanks very much for your comment.

All statistical analyses were performed using GraphPad Prism 9.0. Datas are expressed as the mean ± standard deviation (SD). Comparisons between two groups were performed using Student's t-test and comparisons among multiple groups were performed using one-way ANOVA followed by Tukey's post hoc test.Statistically different at p<0.05,statistically significant at p<0.01.All data used in statistical analyses were obtained from three or more independent repeated experiments.According to your suggestions, we have carefully checked the data statistics involved in the whole paper again to ensure that the significance level is expressed correctly, and also checked all descriptions of the results in the paper in detail to ensure that the interpretation of the results is accurate.

3.Consistency and Reproducibility of Results: Provide additional information about the replicability of the experiments to ensure the reliability and consistency of the results. Describe the details of the control group setup to aid other researchers in replicating the experiment.

Response:

Thanks very much for your comment.

According to your suggestions, we have re-checked and included the detailed information of all instruments and reagents used in the study, and improved the descriptions of different group settings and different drug administration schemes, which are specifically reflected in the Materials and Methods section.

4.Study Limitations: Discuss the limitations of the study, including factors that might affect the conclusions.

Response:

Thanks very much for your comment.

We have included a detailed discussion on the limitations of the study and the factors that may affect the observed results in the Discussion section. All specific changes have been highlighted on Page 29-30 line 627-659.

5.Logical Structure: Ensure the paper's clarity and logical flow from the introduction through methods, results, and discussion sections. Provide additional background information on DOX-induced cardiotoxicity to help readers understand its significance and the need for treatment approaches. Enhance the interpretation of experimental results and their relevance. In the discussion, provide future research directions and prospects for potential clinical applications.

Response:

Thanks very much for your comment.

We have reviewed and revised all parts of the article and adjusted the structural order of some paragraphs to ensure that the logical structure is clear, correct and complete.In addition, we improved the relevant background of DOX-induced cardiac toxicity in the Introduction and Discussion sections. Its high incidence in clinical treatment and poor prognosis do not only affect the therapeutic dose of DOX, but also affect the quality of life of cancer survivors and even shorten their life expectancy.The only FDA-approved drug that can be used to treat DOX-induced cardiotoxicity, dexrazoxane, exhibits side effects.Therefore, it is cardinal to find a drug that can improve DOX cardiotoxicity, explore its mechanism of action, and translate it into clinical application.

We also re-examined and refined the interpretation of the experimental results and their correlation. In the Discussion, we put forward our thoughts on the future research directions and prospects of potential clinical application. All specific changes have been highlighted.

6.Citations and Background Research: Update the references to include the latest and relevant studies to support the research hypothesis and conclusions. Clearly state the motivation and research questions behind the study, and provide background research to illustrate the importance and relevance of the study.

Response:

Thanks very much for your comment.

According to your suggestions, we have revised the Introduction and Discussion sections to add relevant researches from recent years to better support our hypothesis and conclusion. Explanations for the importance and relevance of the research context have also been strengthened. All specific changes have been highlighted.

7.Ethical Compliance: Given the involvement of animal experiments, ensure compliance with all relevant ethical guidelines and clearly state the ethical approval number in the paper.

Response:

Thanks very much for your comment.

The study was approved by the Institutional Committee for the Protection and Utilization of Animals of Jilin University(Approval Number:2023 No. 463).All handling of laboratory animals during experiments was in accordance with the Guidelines for the Management and Use of Laboratory Animals published by the National Institutes of Health. Animal studies were conducted in accordance with ARRIVE guidelines. We have included this in the Animals and treatment section.

[References]

1、Jensen RA,Acton EM,Peters H.Doxorubicin cardiotoxicity in the rat: comparison of electrocardiogram,transmembrane potential,and structuraleffect [J].J Cardiovasc Pharmacol,1984,6(1):186-200.

Yi X,Bekeredjian R,DeFilippis NJ,et al.Transcriptional analysis of doxorubicin-induced cardiotoxicity [J].Am J Physiol Heart Cir Physiol, 2006, 290(3):H1098-H1102.doi：10.1152/ajpheart.00832.2005. 

5、Szczesny B, Módis K, Yanag K,et al. AP39 [10-oxo-10-(4-(3-thioxo-3H-1,2 -dithiol-5yl)phenoxy)decyl) triphenylphosphonium bromide], a mitochondrially targeted hydrogen sulfide donor, stimulates cellular bioenergetics, exerts cytoprotective effects and protects against the loss of mitochondrial DNA integrity in oxidatively stressed endothelial cells in vitro.Nitric Oxide. 2014 September 15; 41: 120–130. doi:10.1016/j.niox. 2014.04.008.

In all, we appreciate for your warm work earnestly, and we revised our paper point-by-point. Once again, thank you very much for your comments and suggestions. 

Kind Regards, 

Bin Zhang,Yangxue Li,Ning Liu and Bin Liu

---

## [Decision Letter · Decision Letter 1]

18 Feb 2024

PONE-D-23-34322R1AP39 ameliorates doxorubicin-induced cardiotoxicity by regulating the AMPK/UCP2 pathwayPLOS ONE

Dear Dr. Liu,

Thank you for submitting your manuscript to PLOS ONE. After careful consideration, we feel that it has merit but does not fully meet PLOS ONE’s publication criteria as it currently stands. Therefore, we invite you to submit a revised version of the manuscript that addresses the points raised during the review process.

We look forward to receiving your revised manuscript.

Kind regards,

Mohamed Abdel-Daim, Ph.D.

Academic Editor

PLOS ONE

Journal Requirements:

Reviewers' comments:

Reviewer's Responses to Questions

**Comments to the Author**

1. If the authors have adequately addressed your comments raised in a previous round of review and you feel that this manuscript is now acceptable for publication, you may indicate that here to bypass the “Comments to the Author” section, enter your conflict of interest statement in the “Confidential to Editor” section, and submit your "Accept" recommendation.

Reviewer #2: (No Response)

Reviewer #3: (No Response)

2. Is the manuscript technically sound, and do the data support the conclusions?

Reviewer #2: (No Response)

Reviewer #3: Yes

3. Has the statistical analysis been performed appropriately and rigorously? 

Reviewer #2: (No Response)

Reviewer #3: Yes

4. Have the authors made all data underlying the findings in their manuscript fully available?

Reviewer #2: (No Response)

Reviewer #3: Yes

5. Is the manuscript presented in an intelligible fashion and written in standard English?

Reviewer #2: (No Response)

Reviewer #3: Yes

6. Review Comments to the Author

Reviewer #2: (No Response)

Reviewer #3: I suggest title modification to explain what is AP39?

The suggested title will be as follows.

AP39, a novel mitochondria-targeted hydrogen sulfide donor ameliorates doxorubicin-induced cardiotoxicity by regulating the AMPK/UCP2 pathway.

Pathogenesis of DOX cardiotoxicity should be covered in depth in introduction and discussion sections with reference to the possible mechanisms of potential cardioprotective agents. The following references might be helpful.

https://doi.org/10.1016/j.biopha.2017.04.033

https://doi.org/10.1007/s00280-017-3413-7

https://doi.org/10.3389/fphar.2019.00635

7. PLOS authors have the option to publish the peer review history of their article (what does this mean?). If published, this will include your full peer review and any attached files.

Reviewer #2: No

Reviewer #3: **Yes: **ZEINAB MAHASNEH

---

## [Author Response · Author response to Decision Letter 1]

20 Feb 2024

Dear editor and dear reviewers: 

Thank you for your letter and the reviewers’ comments regarding our manuscript titled “AP39 ameliorates doxorubicin-induced cardiotoxicity by regulating the AMPK/UCP2 pathway” (ID: PONE-D-23-34322R1). The comments were all valuable and very helpful in amending and improving our paper, as well as our research. We have addressed all the comments and have made corrections to the manuscript (highlighted in yellow).We hope that the revised manuscript now meets the requirements of PLOS ONE. Our point-by-point responses to all comments are presented below:

Reviewer #3:

1.I suggest title modification to explain what is AP39?

Response: 

Thank you very much for your comment. 

We have changed the title to ”AP39, a novel mitochondria-targeted hydrogen sulfide donor ameliorates doxorubicin-induced cardiotoxicity by regulating the AMPK/UCP2 pathway”as you suggested.

2. Pathogenesis of DOX cardiotoxicity should be covered in depth in introduction and discussion sections with reference to the possible mechanisms of potential cardioprotective agents. The following references might be helpful.

https://doi.org/10.1016/j.biopha.2017.04.033

https://doi.org/10.1007/s00280-017-3413-7

https://doi.org/10.3389/fphar.2019.00635

Response: 

Thank you very much for your comment.

 Based on your suggestions and regarding the literature you provided, we have further discussed and improved the information concerning the pathogenesis of DOX-induced cardiotoxicity in the Introduction and Discussion sections,referencing the possible mechanisms of potential cardioprotective agents. All specific changes have been highlighted.

We appreciate your earnest work reviewing our manuscript, which we have revised point-by-point per your suggestions. Once again, thank you very much for your comments and suggestions. 

Kind Regards, 

Bin Zhang,Yangxue Li,Ning Liu and Bin Liu

---

## [Editor Report · Decision Letter 2]

26 Feb 2024

AP39, a novel mitochondria-targeted hydrogen sulfide donor ameliorates doxorubicin-induced cardiotoxicity by regulating the AMPK/UCP2 pathway

PONE-D-23-34322R2

Dear Dr. Liu,

We’re pleased to inform you that your manuscript has been judged scientifically suitable for publication and will be formally accepted for publication once it meets all outstanding technical requirements.

Kind regards,

Mohamed Abdel-Daim, Ph.D.

Academic Editor

PLOS ONE
---

## [Editor Report · Acceptance letter]

21 Mar 2024

PONE-D-23-34322R2 

PLOS ONE

Dear Dr. Liu, 

I'm pleased to inform you that your manuscript has been deemed suitable for publication in PLOS ONE. Congratulations! Your manuscript is now being handed over to our production team.

Kind regards, 

on behalf of

Professor Mohamed Abdel-Daim 

Academic Editor

PLOS ONE